# A Qualitative Analysis of Black and White Adolescents’ Perceptions of and Responses to Racially Targeted Food and Drink Commercials on Television

**DOI:** 10.3390/ijerph182111563

**Published:** 2021-11-03

**Authors:** Alysa Miller, Omni Cassidy, Tenay Greene, Josh Arshonsky, Stephanie L. Albert, Marie A. Bragg

**Affiliations:** 1Department of Psychology, University of Illinois at Chicago, 1007 W. Harrison St., 1059 BSB, Chicago, IL 60607, USA; alysamill@gmail.com; 2Department of Population Health, New York University (NYU) Grossman School of Medicine, 180 Madison Ave., New York, NY 10016, USA; Omni.Cassidy@nyulangone.org (O.C.); tag359@nyu.edu (T.G.); Joshua.Arshonsky@nyulangone.org (J.A.); Stephanie.Albert@nyulangone.org (S.L.A.); 3Department of Public Health Nutrition, NYU School of Global Public Health, 708 Broadway, New York, NY 10003, USA

**Keywords:** obesity, targeted marketing, adolescents, racial health disparities

## Abstract

Food and beverage marketing is a major driver of childhood obesity, and companies target their least nutritious products to Black youth. However, little is known about adolescents’ perceptions of and responses to racially targeted food marketing. In this qualitative study, we investigated how Black and White adolescents perceived and responded to racially targeted television commercials for food and beverages. We recruited 39 adolescents aged 12–17 years in New York City to watch a series of commercials and then participate in an in-depth interview using a semi-structured interview guide. The research team recorded, transcribed, and analyzed interviews using ATLAS.ti. Overall, participants responded positively to commercials that featured celebrities. They were also able to recognize the commercials and reported they had been exposed to marketing from these companies on social media and in subways/buses. Many participants considered the advertised brands as healthy or able to enhance athletic performance because of their endorsement by or association with athletes. Participants also understood that marketers were using racial targeting in their ads but that targeting did not translate into improved perceptions or responses towards advertised products. These findings suggest the need to empirically evaluate and further explore Black and White adolescents’ responses to racially targeted food marketing.

## 1. Introduction

High rates of adolescent obesity persist and disproportionately impact minority youth in the United States [1]. Approximately 28% of Black adolescents (12–19 years old) meet criteria for obesity compared to 16% of White adolescents [1]. Adolescents with obesity are more likely to remain obese into adulthood [2] and develop chronic diseases, such as cardiovascular disease and type 2 diabetes [3].

One powerful environmental risk factor for high obesity rates is the pervasive prevalence of food and beverage marketing. The food and beverage industry spends USD 1.8 billion annually on marketing to youth [4]. Between 2013 and 2017, the industry increased spending on TV advertisements (“ads”) targeting Black communities by more than 50% [5]. Ad expenditures translate to Black adolescents, in particular, being exposed to an estimated 6200 food and beverage TV ads each year [5]. Black youth are also targeted with more ads for foods and beverages of poorer nutritional value as compared to White adolescents [6,7,8]. One report in 2015 showed that Black adolescents were exposed to more than twice as many ads for snack foods on TV than White adolescents, an increase of more than 80% compared to previous years [7]. Exposure to ads can influence preferences [9], self-reported intake [10], and consumption [10], contributing to poor diet and obesity-related health disparities [11].

Racially targeted marketing refers to a fundamental marketing strategy companies use in which they promote their products to specific demographic groups [12]. Despite this routine marketing practice, existing data about the targeted marketing of unhealthy food and beverages to Black youth are scarce. Existing available data show that youth of color are subject to disproportionately large amounts of unhealthy food advertising on television, billboards, and other traditional media relative to the general public [5,11]. According to a 2019 report by the Rudd Center for Food Policy & Obesity, Black children and teens saw 90% more ads for snacks and sugary drinks on TV compared to White youth [5]. Much of racially targeted marketing occurs in-person and in digital settings frequently visited by adolescents. Although social marketing has been used to advance healthy public health programs (e.g., participation in food assistance programs) [13], less is known about food companies’ promotion of unhealthy foods that may exacerbate health disparities.

Food and beverage companies target Black consumers using many of the defining features of social marketing. These include developing tailored marketing campaigns that reach Black consumers repeatedly from different channels [14]; that induce emotional responses [15]—such as humor or happiness—to increase consumer engagement via happiness management [14,16,17,18]; that use different forms of endorsement from celebrities and peers to create trust and credibility; and that convey culturally relevant and inclusive themes [19,20,21,22], linking Black consumers to common experiences or traditions.

Such marketing techniques have several psychological and behavioral implications that can influence purchases. Ads featuring culturally relevant and inclusive themes, for instance, can reinforce a sense of identity with young Black consumers [19,23]. Such ads can also lead to favorable responses in the targeted consumer group [21,22,24]. Ads with Black spokespersons may also influence purchasing behaviors. Research suggests that Black consumers are more likely to purchase brands with Black spokespersons compared to White spokespersons [25,26]. However, little is known about how Black adolescents perceive racially targeted marketing.

Targeted marketing is a common industry practice, but when such marketing disproportionately promotes unhealthy foods, it may contribute to health disparities that affect Black communities. Countering these marketing practices, then, is an important strategy to promote public health and shrink racial health disparities. As it turns out, however, this countering strategy is complex and extremely challenging to both research and put into practice [27]. Food itself is difficult to classify as harmful or toxic; it is not the same application as for tobacco products, for example. Furthermore, food marketing is expansive and pervasive, driven by profit. Food advertising and promotion is also protected by law under the First Amendment [28]. To date, most public health arguments for restricting the marketing of sugary drinks and unhealthy foods are made on the basis of age or developmental stage—restricting marketing to a segment of the population (i.e., children) who are unable to fairly interpret such marketing [20]. However, arguments for restricting marketing by race or ethnicity are complex. On one hand, promoting unhealthy food products to communities who face high rates of diet-related diseases warrants interventions. On the other hand, restricting targeted marketing can inadvertently convey that certain communities of color are somehow less able to fend for themselves in the marketplace compared to other communities [20]. With these considerations in mind, the current strategy from a public health perspective is to mobilize communities affected by the marketing to respond with resistance and advocate for changes in marketing practices.

Previous research on this approach has sought to understand Black consumers’ perceptions and responses to targeted marketing so as to inform appropriate and effective counter-marketing strategies [23,29]. In a 2017 qualitative study, researchers conducted focus groups with Black adults and youth to understand their views of targeted marketing, including its relevance as a social justice issue. Researchers conducted two focus groups with participants, but before the second focus group, they provided participants with an informational booklet about targeted marketing. Findings revealed varied responses among both Black adults and youth. Some participants viewed marketing practices as expected and understandable, because they aligned with celebrated business practices of earning the most profit possible [23]. Other participants discussed the role of individual responsibility in determining how detrimental or influential food and beverage marketing tactics were, asserting that consumers needed to choose better themselves [23]. A more recent qualitative assessment explored Black and Latino adolescents’ attitudes toward food and beverage brands and marketing that targets them directly. Findings showed that overall, participants indicated more positive attitudes about the targeted brands than the non-targeted brands, and participants often described the targeted brands discussed in the focus groups as being for someone most like them [29]. Some youth expressed appreciation for marketing that recognized “someone like them” as potential consumers, while others raised ethical concerns about the targeting of unhealthy foods [29].

However, few studies examine how adolescents perceive and respond to racially targeted food and beverage advertisements [22,30,31,32,33,34,35] on television immediately after being exposed to them. This qualitative pilot study aimed to fill that gap by exploring how Black and White adolescents perceive and respond to racially targeted television commercials for food and beverages. Specifically, we aimed to characterize how factors such as targeting, celebrities, ad recognition, health claims, and socioeconomic and neighborhood factors influence perceptions and reactions to ads among the adolescents participating in the study.

## 2. Materials and Methods

The NYU Grossman School of Medicine (NYUSOM) Institutional Review Board approved all study protocols and procedures.

### 2.1. Study Design

This study employed a qualitative design in which in-depth key informant interviews were conducted to understand how factors influence perceptions and reactions to ads. Qualitative research methodologies are often the most appropriate study designs to use when research is in an exploratory or formative stage and little is known about a phenomenon [36,37]. Because no food marketing studies have included both Black and White adolescents for qualitative interviews, this investigation functions as an exploratory study to begin to characterize the extent to which Black and White adolescents hold similar or differing views on racially targeted marketing. Exploratory studies are able to help generate new hypotheses for future studies and provide data that can inform the design of more robust experimental studies with larger samples [38,39].

### 2.2. Identifying Food and Beverage Commercials

To select television commercials, the research team used the methodology described in Bragg et al. (2019) [40]. Research assistants searched the Kantar Media database on AdScope to identify matching Black- and White-targeted food and beverage advertisements. We considered ads as matching if they featured the same brands and were within 15 s in length of the counterpart ad. The selected television advertisements all featured adolescents, originated from companies with the highest advertising expenditures, and aired in 2010 or later. The research team purchased a set of 16 advertisements (8 Black-targeted; 8 White-targeted) to use for the study, representing 8 brands: Trolli Gummy Candy, Skittles, Pebbles Cereal, Reese’s Puffs Cereal, Panera Bread, Gatorade, Pepsi, and McDonald’s (Table 1).

### 2.3. Sampling and Recruitment

Using a convenience sampling approach, research assistants recruited participants ages 12–17 years living in New York City—specifically those living in Manhattan, Brooklyn, the Bronx, or Queens—from public parks and local community organizations, as well as outside of local schools. Because our research team lives and works in New York City, we used convenience sampling to recruit adolescents who lived within the area. Potential participants completed a screening questionnaire to determine their eligibility. All participants had to be between the ages of 12 and 17, identify as either Black/African American or White, and have parental consent to participate. Participants also had to provide assent. Because recruitment took place over an extended period, the study team was able to end enrollment when saturation was reached rather than identifying a sample size a priori [41]. In total, we recruited 39 adolescents (20 Black, 19 White) to participate in the study.

### 2.4. Data Collection and Management

The principal investigator (M.B.) developed a semi-structured interview guide to help interviewers facilitate discussion after participants watched each commercial. The interview guide was based on prior research—Spear and Singh’s (2004) attitude toward the brand measure and Madden, Allen, and Twible’s (1988) attitude toward the ad inventory—as well as our research aims [42,43]. Questions assessed adolescents’ responses to ads, their brand engagement, purchasing and consumption behavior, and their recognition of the targeted audience (Table 2). We pilot tested the interview guide and made minor revisions to the guide to improve clarity and flow. The pilot phase also served as an opportunity to train three interviewers.

Interviews were conducted between July 2016 and April 2017. The majority of interviews (*n* = 36) were conducted in-person immediately after obtaining consent and assent at the recruitment site (i.e., park, community organization, school). When it was not possible to obtain parental consent onsite or when the participant did not have sufficient time to complete the interview, the research team scheduled an in-person interview at the NYUGSOM office (*n* = 3).

Each interview began with a brief overview of the purpose and procedures of the study. Participants also had an opportunity to ask questions before the interview began. Then, each participant watched a total of six commercials. Participants randomly viewed either only Black-targeted or only White-targeted commercials. Randomization to commercial condition was independent of participants’ self-identified race. That is, a Black youth could be randomized to view the White-targeted commercials, and vice versa. The specific order and set of commercials each participant saw were also randomized. Using an iPad, research assistants showed each commercial individually to participants. Following each commercial, the primary interviewer paused to ask the participant a series of open-ended questions related to the content of the ad and its featured product (e.g., Please describe how you felt when watching the ad; What groups of people or types of consumers do you think the company was trying to reach with this ad and why?)

The primary interviewers conducted each interview, with a second interviewer present to administer the prescreening survey, take notes, and ask clarifying questions. Interviews lasted approximately 30 min and adolescent participants received USD 15 for their time participating in the interview. All interviews were audio-recorded with permission and were professionally transcribed, reviewed for accuracy, and de-identified. Transcripts were entered into ATLAS.ti for data management, coding, and analysis.

### 2.5. Data Analysis

The principal investigator (M.B.) and research assistants analyzed the data. The first stage of analyses included reviewing each of the transcripts and discussing the interviews. The researchers then used a deductive approach (Hyde, 2000) to collectively identify, analyze, and define codes, supporting themes, and relevant relationships through the content of the interviews themselves [44]. Each theme was defined in a coding manual and assigned nodes in ATLAS.ti for coding. Research assistants conducted three rounds of reliability testing with sample transcripts, and determined inter-coder reliability based on percent agreement of at least 80% for all codes per transcript. When agreement was less than 80%, coders discussed the discrepancies and resolved them using consensus and by refining and merging codes, if needed. The team then divided the remaining transcripts (*n* = 36) and independently analyzed the interviews using the coding manual. At the conclusion of the coding, the research team met to review the data and discuss emerging themes and findings.

In-depth interviews utilized the framework provided by Schwandt, Lincoln, and Guba (2007) on credibility, transferability, dependability, and confirmability [45]. To check the validity of interviewers, research assistants used “peer debriefing” to develop and test the emerging themes of interviews. Peers were members of the study team who were not involved in data collection, and could therefore provide a more objective outside view for honing hypotheses. For transferability, the research team transcribed themes related to the rich descriptive data so that the narrative of the data could be used by others. Rich quotes were particularly valuable for generating the narrative. For dependability and confirmability, one author who was not involved in data collection (J.A.) provided an external audit by reviewing the data and reconstructing the process results. Taken together, these steps enabled the study team to gain additional confidence in the validity of the interviews.

## 3. Results

### 3.1. Study Participants

Table 3 summarizes participant sociodemographic data. We interviewed 46 participants and excluded seven participants due to incomplete data (e.g., interviews not completed because of inclement weather at interview site), resulting in a final analytic sample of 39 participants. Twenty participants (51.3%) identified as Black and nineteen (48.7%) identified as White. Our sample consisted of 18 participants who identified as male, and 21 participants who identified as female. The average age of participants was 14.18 (SD = 1.62). Ten Black participants in our sample viewed Black-targeted commercials, while another ten Black participants viewed White-targeted commercials. Among the White participants, ten viewed Black-targeted commercials, while nine viewed White-targeted commercials.

### 3.2. Qualitative Results

Overall, most adolescents recognized the commercials shown to them. Some participants were aware of the racially targeted marketing but more often discussed targeted marketing in terms of age and activities or interests. Participants reported responding positively to the commercials in our sample, and all of them had been previously exposed to the commercials on social media. They reported particularly enjoying the ads that featured celebrities. However, adolescents expressed mixed attitudes toward the products promoted in the commercials. Key themes that emerged from the semi-structured interviews are highlighted in the following sections.

### 3.3. Ad Targeting

After watching all six commercials, when asked whether they thought the companies shown were trying to attract a particular group of consumers, adolescents’ responses varied. Respondents felt that companies were targeting their audience by race or ethnicity, age group, or activity and interests.
“Well, I guess for the cereal and the gummy bears, and the McDonald’s, also the Gatorade and the Pepsi, maybe just like youth. Maybe like kids to their early twenties.”“Well, by the people they are promoting they are all white. So I would say White but I wouldn’t say that’s their main intention.”

Among all interviewees, 32 adolescents indicated that—after prompts from interviewers—the ads they had seen targeted consumers by race or ethnicity (see Table 4). Seventeen of the Black adolescents correctly identified the targeted racial group in their respective ad conditions, and 14 White adolescents correctly identified the targeted racial group in their respective ad conditions.

Before even being asked about the targeted consumer group at the end of the interview, some adolescents alluded to targeting in specific commercials. Elaborating on the reason for feeling “agitated” by the Black-targeted Reese’s Puff commercial, an adolescent said:“It’s kind of overplayed and you can see like that it’s geared towards a certain audience with the kid and the rapping, but it’s not really working out so well for them.”

Many adolescents discussed the youth-targeting qualities of the advertisements, often distinguishing their age group from younger children.
“I liked that [the ad for Panera Bread] was appealing to a young demographic. This place has tons of good food, and you can have fun with your friends.”“[It] kind of showed an aspect of how teenagers acted, and it was appealing to people my age.”“I use to eat [McDonald’s] in kindergarten, but my friends and I don’t anymore because it’s just really unhealthy for you.”

For adolescents, race and age were not the only markers of targeting. Many adolescents, when asked whether they would share the ad with others, discussed sharing the ads with friends or family whose activities or values aligned with the brand images displayed in those ads. Participants, for example, would share ads or purchase advertised products for “friends who play sports” or “people that also watch” shows featured in the ads (i.e., The Bachelor). One participant mentioned they would not share the McDonald’s commercial with friends or family because:“They’re all about sports, they don’t really do TV stuff like that.”

### 3.4. Responses to Ads

In all ad conditions, participants’ responses to each commercial were influenced by the celebrities featured in the commercials, as well as previous exposure to commercials for the same product or brand.

When discussing what they thought of the commercials, participants’ attitudes were frequently shaped by the presence of celebrities. Of the 16 food and drink commercials in our sample, nine included sports or television celebrities who teens recognized and commented on (Table 1). Participants who saw Black-targeted ads mentioned the featured Black celebrities by name 85 times, while participants who saw White-targeted ads recognized and mentioned the featured White celebrities by name 68 times.

Adolescents discussed the celebrities as a factor for enjoying the ad, connected the qualities of the actors to the brands they were advertising, and factored them into their decision to share the ad with others.
“Usain Bolt stands out because he is a good person.”“I actually kind of liked [the commercial], because it had James Harden who’s a big celebrity.”“I liked [the commercial], because I watch The Bachelor, and I know that was Ben…I actually hate the guy, but that’s beside the point. But yeah I liked them, because I’ve seen them before.”“[Marshawn] Lynch likes Skittles and he’s cool and a lot of people like him. So maybe that’s just a reason you should like Skittles.”

Adolescents indicated that they had previously seen commercials of the same brand or for the same products in various physical and digital spaces. Participants mentioned seeing or hearing the advertised brands on social media 128 times, on television 256 times, on the radio 6 times, and on other media forms 10 times.

Television was the primary medium through which participants had seen advertisements. Two participants mentioned having seen brands on children’s television shows, specifically.
“I used to see a lot of [Trolli commercials] when I was younger. I used to watch TV a lot more.”“I’ve seen ads for Coco and Fruity Pebbles on TV kids’ shows.”

### 3.5. Purchasing and Eating Behaviors

Participants talked about purchasing and buying the advertised products around key themes of healthfulness, enhancing sports performance, and socioeconomic and neighborhood influences.

In describing whether they’ve had or would purchase the advertised products or share them with others, most participants discussed these intentions largely based on their perceptions of the products as healthy or unhealthy.
“…It’s not the healthiest choice, so I wouldn’t necessarily tell a family member about it.”“They were really good. They were very sweet and nutritious.”“At first, I really liked it because I was in middle school and was like, ‘Oh, unhealthy? Cool!’ But then after seeing this video in health, I don’t think I had it ever again.”“If you eat this cereal, you will be athletic and healthy.”“Yeah, I just can’t eat [McDonald’s] that much because it’s fast food. And I heard there’s rat meat, like 60% mystery meat.”“I like it, but I heard it causes kidney stones and all that.”

Panera was nearly always discussed as healthy or nutritious, whether it was the purpose of the ad or products themselves. However, one participant recognized that not all Panera products are healthy.

“I don’t really like salads, they’re slightly better than they used to be. I don’t know how they’re faking that Panera is healthy. I would know because I get their—what is it—the sugary bagel that’s pretty much a donut.”

Some specific products, such as Gatorade, were discussed within the larger context of health but as enhancing sports performance, specifically.
“Yeah I hear [Gatorade] helps replenish electrolytes.”“[The main message of the ad] is that if you drink Gatorade you could go to like Major League Baseball or something.”“[Gatorade] helps you get energy for the game.”“[Gatorade] fulfills all your exercise needs.”

Gatorade was not the only product perceived as helpful to athletes. One participant noted:“Skittles are proven to enhance sports performance and reaction time. They did a sports science with Marshawn Lynch on it doing a bunch of tests, and every single one he improved.”

Socioeconomic and neighborhood factors were also raised when adolescents discussed the availability of the advertised products in their own communities. Participants described seeing the brands’ products in local drugstores, supermarkets, vending machines, and even food trucks. When describing their communities and neighborhoods, adolescents almost always identified the ubiquity of certain brands and products—the ease of accessing them in their communities. Others, however, recognized the differences in availability of products between communities.
“Considering I live in America and in a city, yes [McDonald’s is part of my community].”“You walk into any store like CVS and it’s just there.”“You’ll see them when people are grocery shopping.”“I guess...it’s in all of the vending machines and in all the stores.”“At the food trucks every day at lunch where I get lunch they like sell them at each food truck and they are like the number one candy people buy.”“There’s a McDonald’s a few minutes away, and I know 474 people who go to it.”“…I can walk around the corner and then up a few blocks and McDonald’s right there.”

Some participants recognized that the availability of certain brands and restaurants differs between low-income communities and high-income ones. Adolescent participants tended to describe those differences in terms of what they considered healthy.
“[Panera is not part of my community] cause I live in the projects.”“[It’s not part of my community] because we up in the hood. And they don’t really have supermarkets here.”“I seen commercials [for Panera] but never, ever went there, or seen one.”Two adolescents also commented on the role of food and drink companies in influencing the availability of brands in different neighborhoods.”“No because I haven’t really seen Skittles make an effort to be a part of our community and I don’t really think it does in the first place.”“Yes, Gatorade does support a lot of high school sports and that’s the thing everyone drinks. And that’s where they sponsor most of the time.”

## 4. Discussion

The present study provides insight into how Black and White adolescents perceive and respond to racially targeted television commercials for food and beverages. Our exploratory findings demonstrate that most adolescent participants recognized the racial targeting qualities of the commercials shown to them after being prompted by interviewers, but unprompted, many participants discussed targeted marketing mostly in terms of age and interests. When asked whether they liked the selection of commercials, the majority of adolescents responded positively to and enjoyed commercials that featured celebrities. Participants also discussed having previously seen the commercials in our study on social media platforms, such as YouTube. Many participants considered the advertised brands as healthy because of their endorsement by or association with athletes. Overall, participants recognized the targeted nature of advertising, reported positive responses to the celebrity endorsements featured in ads, and noticed increases in social media advertising of food and beverage brands. These findings demonstrate the pervasive reach and wide recognizability of food advertising among adolescents, and indicate the need for more research focused on adolescents. Additional adolescent-focused research is needed to inform policy discussions on the extent to which youth ages 13–17 years should receive policy protections. Indeed, the majority of food marketing studies have focused on children [46,47], and food marketing policies aim to shield children from ads [48] while older adolescents remain bombarded by ads [49].

### 4.1. Targeting

Participants recognized racial targeting after being prompted by interviewers, but they discussed targeted marketing mostly in terms of age and activities or interests, which differs from prior research [29]. Research has shown that racial similarity, for example, evokes positive effects among the target market. In our study, most participants responded positively to commercials regardless of the race of the spokesperson featured. Perhaps some of the adolescent participants responded positively to commercials that represented their age group and behaviors more than their race. Either way, this finding suggests that effective counter-marketing campaigns should implement a range of targeted marketing tactics, rather than exclusively racial-targeting. More research is needed, though, on what specific features of food and drink advertisements evoke responses from adolescents.

### 4.2. Celebrity Influence

In our study, adolescent participants reported enjoying commercials that featured celebrities. Research shows that the use of celebrity endorsements in marketing can enhance brand loyalty and the desirability of a product, leading consumers to easily recognize brands [50,51,52] and to view brands more positively. This effect is especially powerful among adolescents who idolize celebrities as part of their identity development [53,54] and desire to differentiate themselves from their parents. Our data underscore the powerful influence of celebrity endorsements, regardless of celebrities’ likeability among consumers. Given the persuasive power of celebrities, it may be effective to consider using celebrity spokespersons—athletes, in particular—in future counter-marketing campaigns.

Many participants viewed food and drink brands endorsed by celebrity athletes as healthy, regardless of the brands’ nutritional content. A commercial for Skittles, for example, featured football running back Marshawn Lynch. One adolescent participant who watched this commercial discussed how Skittles improves athletic performance. This perception of Skittles and other unhealthy brands as healthy is not unique to adolescent consumers. A previous study revealed that parents perceive food brands as healthier when they are endorsed by a professional athlete and are more likely to purchase those brands [55]. Food and beverage companies pay professional athletes millions of dollars to endorse predominantly unhealthy brands [56]. Such athlete endorsements often lead consumers to overestimate the healthfulness of such brands.

### 4.3. Shifts in Digital Media

In our study, many participants discussed having seen commercials previously on social media platforms, such as YouTube, Facebook, and Instagram. This finding corroborates current marketing trends in the food and beverage industry. Although TV is the primary marketing medium to target children, the food and beverage industry has shifted their ad spending in favor of digital marketing [57,58], especially through social media platforms such as YouTube. This marketing shift reflects changes in media use among adolescents. In the United States, 95% of adolescents aged 13–17 own a smartphone, and 89% access the internet several times each day [59]. Among adolescents, the use of social media is also widespread. Data collected from Pew Research Center show that over 8 in 10 American adolescents use YouTube while 72% use Instagram, 69% use Snapchat, 51% use Facebook, and 32% use Twitter [59]. The familiarity of and previous exposure to the targeted brands in our sample indicates that social media platforms are important spaces to promote future counter-marketing campaigns for adolescents.

### 4.4. Limitations

Our pilot study has several limitations. First, the sample size is small. A similar study with a larger sample size would add to the existing literature and would enable us to examine whether Black and White adolescents hold significantly different perceptions of racially targeted food marketing that may shape their preferences for products featured in targeted ads. However, it is worth noting that saturation was reached, and new themes related to factors influencing perceptions or reactions to ads did not emerge in the final few interviews conducted. Second, all participants were recruited through public parks, community organizations, and near schools in New York City; as such, the results from this sample may not be generalizable to all adolescents. Third, the fact that all participants resided in New York City may also affect their exposure to food and drink advertisements. Adolescents living in more suburban and rural areas may not have similar ad exposure, perhaps because of less use of public transportation and ad density. Food ad density may be higher in urban areas due to more available ad space on public transportation and city sidewalk kiosks that post ads. Fourth, because this was an exploratory qualitative study designed to evaluate patterns and generate hypotheses rather than to estimate or quantify a problem, the findings may not be generalizable to all Black and White adolescents. Future studies should experimentally examine how celebrity endorsements affect Black and White adolescents’ attitudes and behaviors. Despite these limitations, this pilot study provides insightful information on adolescents’ perceptions and responses to racially targeted food advertisements and communicates the unique perspectives of adolescents.

## 5. Conclusions

Food and beverage companies spend nearly USD 2 billion each year on marketing that pervades the lives of children and adolescents and contributes to childhood obesity. Our findings reinforce that companies use a number of strategies to promote such products, including athlete/celebrity endorsements and social media promotions, which hold unique appeal among adolescents. Such marketing typically promotes unhealthy food and drinks, so it is imperative to understand how adolescents—especially adolescents of color—perceive and respond to food advertising in order to develop effective strategies to combat advertisements’ negative impacts on diet and health. In our small sample, Black and White adolescents shared similar perceptions of targeted ads, reinforcing the broad appeal of advertising.

This qualitative study suggests that adolescents may recognize the racial targeting qualities of different advertisements, but recognize targeting mostly in terms of age and interests. The study also suggests that adolescents are highly interested in food and beverage advertisements that include celebrity endorsements, and may be persuaded by athlete endorsements that suggest sugary sports drinks are healthy. Findings provide additional evidence about the potentially harmful effects of targeting adolescents with unhealthy food and beverage ads. Combating the negative effects of unhealthy food and drink promotions requires a multi-pronged approach that involves equipping communities with improved health knowledge, as well as informing regulations that may reduce or eliminate such marketing to vulnerable groups. Data, for example, may inform the development of counter-marketing strategies that incorporate celebrity or athlete appeal while also aligning with healthful promotion in this new social media marketing frontier. Although the issues around athlete and celebrity endorsements are complex, partnering with athletes or celebrities to discuss the ways they may support these efforts and raising their awareness around the effects of such endorsements may hold promise. Continuing to gather data in these areas will further support the ability to improve food environments for communities of color. Future studies should include a larger sample of adolescents, including adolescents of other racial/ethnic groups who are targeted in food ads (e.g., Latinx and Asian adolescents). In fact, larger qualitative and quantitative studies that include a large sample of diverse adolescents may generate such data on perceptions and effects of targeted marketing across a multicultural sample. Overall, these findings suggest that adolescents may benefit from counter-marketing and advocacy efforts to reduce their exposure to targeted food and beverage ads.

## Figures and Tables

**Table 1 ijerph-18-11563-t001:** Description of ad stimuli and featured celebrities.

Ad	Brand	Celebrities	Ad Condition	Ad Description
1	Gatorade	Usain Bolt	Black-targeted	Bolt appears to run a 40 m dash, then hydrates with Gatorade afterward
2	McDonald’s	None	Black-targeted	TV spot for McDonald’s Money Monopoly game, showing people eating McDonalds and winning the game
3	Trolli	James Harden	Black-targeted	Harden and student sitting on bleachers in school gymnasium as cats emerge from student’s hair and Harden’s beard
4	Fruity Pebbles	Kyrie Irving	Black-targeted	On the basketball court with fans cheering, Irving—the captain of Team Cocoa—claims that Cocoa Pebbles is his favorite part of breakfast
5	Pepsi	Jamal Lyon	Black-targeted	Lyon enters subway car in NYC; a subway rider notices him and throws him a Pepsi, then everyone in the car starts dancing with Lyon
6	Reese’s Puffs	None	Black-targeted	Student in letterman jacket raps about why Reese’s Puffs is the best part of his breakfast
7	Skittles	Marshawn Lynch	Black-targeted	Lynch in the weight room doing strength training with dumbbells and other equipment made of Skittles
8	Panera Bread	None	Black-targeted	Group of teenage friends gathered around a table eating Panera’s new Power Chicken Caesar Salad
9	Gatorade	Bryce Harper	White-targeted	Harper hits a homerun while oozing Gatorade and spreading its mist across the baseball diamond
10	McDonald’s	Contestant from The Bachelor	White-targeted	Recent Bachelor is at McDonald’s checkout counter, making a decision between eating breakfast or lunch
11	Trolli	None	White-targeted	Younger kid plays with his new dog—entirely made of Trolli Sour Crawlers—until an older kid approaches him in his front yard and grabs a handful of the dog and eats it
12	Fruity Pebbles	John Cena	White-targeted	John Cena eating Fruit Pebbles at table and showing off the cereal box in an animated wrestling ring
13	Pepsi	Lionel Messi; Didier Drogba; Fernando Torres; Frank Lampard; Sergio Aguëro; and Jack Wilshere	White-targeted	Elite soccer players crowd surf at a concert in order to get their hands on a Pepsi from the distant vending machine
14	Reese’s Puffs	None	White-targeted	Teenager listening to rap about Reese’s Puffs and pretending to DJ during breakfast
15	Skittles	None	White-targeted	Two teenagers standing under stadium bleachers; one admits to the other that he has the “Skittle Pox”, and the other responds by eating one of the Skittle pox and asking whether it’s contagious
16	Panera Bread	None	White-targeted	Group of teenage friends sitting around table at Panera, eating sandwiches and soup while watching a video on one of the friend’s phones

**Table 2 ijerph-18-11563-t002:** Main questions used to facilitate discussions after participants watched TV commercials.

Response to the ad	Participants indicated whether they liked the ad; how the ad made them feel; how many times they had seen the ad within the previous two weeks; where they typically see ads for the advertised product or brand (e.g., on a billboard, TV, in stores that sell food); if the actors in the commercial stood out to them for any reason; and how much they liked the actors in the ad.
Brand engagement	We asked participants if they had previously seen the ad on any social media platforms, websites, or apps such as Twitter, Instagram, or Facebook. Responses to this question were grouped across platforms and sites.
Purchasing and consumption intentions/behavior	Following each ad, participants indicated whether they had ever purchased or consumed the advertised product; how likely they were to purchase the product or have someone purchase it for them; and how likely they were to recommend a friend or family member to purchase the product for themselves or someone else.
Ad targeting	After viewing all six commercials in their condition, research assistants asked participants if—across brands and commercials—they thought the companies were trying to attract a particular group of consumers (i.e., age or racial/ethnic group).

**Table 3 ijerph-18-11563-t003:** Sociodemographic characteristics of the sample (*n* = 39).

Characteristic	Mean (SD) or Percent
Average Age (years)	14.18 (1.62)
Gender	
Female	53.8
Male	46.2
Race	
Black	51.3
White	48.7
Grade	
6th	2.6
7th	17.9
8th	25.6
9th	15.4
10th	20.5
11th	10.3
12th	7.7

**Table 4 ijerph-18-11563-t004:** Frequency of key themes in participants’ responses.

Themes	Black Participants (*n* = 20)		White Participants (*n* = 19)	
	Black-Targeted Ads Condition (*n* = 10)	White-Targeted Ads Condition (*n* = 10)	Black-Targeted Ads Condition (*n* = 10)	White-Targeted Ads Condition (*n* = 9)
Targeting				
*None mentioned*	1	0	0	1
*Youth-targeted*	1	1	1	1
*Black-targeted*	7	0	7	0
*White-targeted*	1	10	2	7
*Diverse targets*	1	0	1	1
Celebrity Influence				
*Mentions celebrity of one race*	6	3	6	6
*Mentions celebrities of multiple races*	3	6	4	2
Ad Recognition				
*Television*	9	11	10	10
*Print/Billboards*	7	10	7	9
*Social Media*	7	9	7	9
*Other (i.e., YouTube, streaming platform)*	3	4	4	8
Health Claims				
*Benefits of consuming product*	8	7	9	9
*Motivates Purchase of Product*	1	1	2	0
*Barrier to Purchasing Product (Unhealthy)*	2	2	7	2
Socioeconomic and Neighborhood Factors				
*Products accessible in community*	8	11	8	8
*Products inaccessible in community*	2	8	4	4
*Observed people buy/consume product*	7	8	9	6
*Did not observe people buy/consume product*	6	5	9	10

## Data Availability

Data can be available upon request. Please contact the corresponding author for further information. The datasets generated and/or analyzed during the current study are not publicly available due to containing identifiable information for participants (i.e., interview transcripts), which are not possible to de-identify, but are available from the corresponding author on reasonable request.

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
