# Peer review of "A Qualitative Analysis of Black and White Adolescents’ Perceptions of and Responses to Racially Targeted Food and Drink Commercials on Television"

_ijerph, 2021, doi:10.3390/ijerph182111563_

Round 1
Reviewer 1 Report
The proposal is attractive. However, to improve, you need the next questions:
-Theoretical framework: to update some references
-Methods: To complete Methodology with a more qualitative effort (Delphi or in-deep interviews, in this case)
-Results: it will be improved with the methodological new tools
-Conclusions: too much brief. You have to ampliate it.
Reviewer 2 Report
Dear authors,
I find the paper very interesting and topical.
Having said that, I would like to make the following observations:
- The introduction should better justify the aim of the research from a racial point of view. It should also include a few lines explaining the reasons why New York was chosen as the field of study.
- It would be useful to include a literature review section. It should define race marketing and its current lines of research. It would also be appropriate to relate it, on the one hand, to other types of marketing, such as social marketing. And on the other hand, with other areas of the social sciences, such as happiness management.
- The methodology section should include a fact sheet on social class in order to find out whether this factor influences young people's food consumption. I also consider the sample to be very small, which is why it is advisable to justify it statistically with bibliographical references.
- In the result section as well as in the discussion, no reference is made to whether the racial effect has an influence on the consumption of food and advertisements.
- Section 4.4, concerning limitations, I would include it in the conclusions. It would also be interesting to expand it.
- The conclusions are scarce. It would be useful to include some lines on practical implications.
- The bibliographical references are very scarce.
I hope that these observations will help you to enrich your paper.
Best regards
Reviewer 3 Report
Dear authors,
Thank you for the opportunity to review the article "A Qualitative Analysis of Black and White Adolescents' Perceptions of and Responses to Racially-Targeted Food and Drink Commercials on Television". The article presents an interesting insight into adolescents' perceptions of food and beverage advertisements. However, I consider it has to be improved in several aspects. Then here are my arguments:
INTRODUCTION
The introduction should be completed by mentioning scientific articles that have studied the subject and explaining a little about its main results. So that the scope of the investigations carried out can be better understood, and the information gaps to be covered by the study in question can be ascertained. For example, lines 80-83 have mentioned some research on marketing strategies that have been carried out, but what are those strategies? Other studies on adolescents' perception regarding food advertisements from a racial perspective are also commented, but what are the conclusions of these studies.
The objective is too general, and it could be specified in more detail as the results are based on different aspects such as targeting, celebrities, ad recognition, health claims and socioeconomic and neighbourhood factors.
METHODOLOGY
Regarding the methodology, it is necessary to comment that a way to validate the interview topics has been commented, based on the verification by two coders of said topics. It does not exclude the possibility of the agreement due to chance, so complementary measures to estimate consistency between evaluations must be used, such as Cohen's Kappa index. In addition, there are other criteria to check the validity of the interviews, for example, adapting the method of Lincoln and Guba (2007) based on credibility, transferability, dependability, and confirmability. Of course, you can use another method, duly scientifically justified, to validate the interview procedure.
Schwandt, T. A., Lincoln, Y. S., & Guba, E. G. (2007). Judging interpretations: But is it rigorous? Trustworthiness and authenticity in naturalistic evaluation. New directions for evaluation, 114, 11-25.
RESULTS AND CONCLUSIONS
According to the study's objective, the results should be compared based on the responses of black-targeted and white-targeted, but the results have not been explained by differentiating these responses.
The conclusions are scarce, do not sufficiently summarize the results.
Round 2
Reviewer 1 Report
The paper has improved a lot. Perhaps you can optimize the interpretation of conclusions, but in general the article sounds well. Thanks for incorporating my suggestions!
Reviewer 2 Report
Dear authors,
The article has been enriched by the revisions made, but I consider that they are not sufficient to accept it for publication.
The article still lacks the following aspects:
- A solid bibliographical reference, the references to social marketing and happiness management are scarce and lack an important corpus of bibliographical references.
- The statistical methodology has strong weaknesses.
- Conclusions lack a corresponding academic justification.
Best regards
Reviewer 3 Report
Thanks to the authors for the reviews made. I would like to clarify that the objective could be a little more summarized: "...To characterize how factors such as targeting, celebrities, ad recognition, health claims and socioeconomic and neighborhood factors influence perceptions and reactions to ads among the adolescents participating in the study"
